# Susceptibility of *Diaphorina citri* to Irradiation with UV-A and UV-B and the Applicability of the Bunsen–Roscoe Reciprocity Law

**DOI:** 10.3390/insects14050445

**Published:** 2023-05-08

**Authors:** Sabina Parajuli, George Andrew Charles Beattie, Paul Holford, Chuping Yang, Yijing Cen

**Affiliations:** 1Citrus Huanglongbing Research Laboratory/Key Laboratory of Bio-Pesticide Innovation and Application/National Key Laboratory of Green Pesticide, South China Agricultural University, Guangzhou 510642, China; parajuli.sabeena25@gmail.com; 2School of Science, University of Western Sydney, Locked Bag 1797, Penrith, NSW 2751, Australia; a.beattie@westernsydney.edu.au (G.A.C.B.); p.holford@westernsydney.edu.au (P.H.); 3College of Electronic Engineering, South China Agricultural University, Guangzhou 510642, China; yangchp@scau.edu.cn

**Keywords:** psyllid, ultraviolet radiation, ED_50_, longevity, reproduction

## Abstract

**Simple Summary:**

The Asian citrus psyllid is the most widely distributed vector of the Asian form of huanglongbing, and the psyllid is a relatively minor pest in the absence of the pathogen. Pesticides used to suppress psyllid infestations can slow the rate of spread of the disease but not prevent it. Psyllid populations are known to be affected by a number of natural factors, including relative humidity, rainfall, ambient temperature, predators, parasitoids, and entomopathogens. However, in a recent study in Bhutan there was little evidence of these factors being responsible for the psyllid rarely occurring at elevations above 1200 m. It was hypothesised that ultraviolet radiation could be the cause. In this study, we observed how UV-A and UV-B affected the different stages of the psyllid. There was a negative impact of UV-A on all stages of psyllids, but UV-B was the more damaging. At the doses given, when adults were treated with UV radiation, there was little effect on the eggs or nymphs produced. Over the range of doses given, the Bunsen–Roscoe law of reciprocity held true for eggs and early instar nymphs. The result of our experiment suggests that ultraviolet light could be a factor affecting the distribution of *D. citri* populations.

**Abstract:**

Populations of *Diaphorina citri* decline with elevation and, in a study in Bhutan, were rarely found above 1200 m ASL. The impact of ultraviolet (UV) radiation, particularly UV-B, on immature stages of the psyllid was proposed as limiting factor. As no studies have been undertaken on the influences of UV radiation on the development of *D. citri*, we examined the effects of UV-A and UV-B on different stadia of the psyllid. In addition, compliance with the Bunsen–Roscoe reciprocity law was examined. Irradiation with UV-A marginally reduced egg hatch and the survival times of emerging nymphs. Early instar nymphs were little affected by this waveband, but the survival of adults was reduced at the higher doses used. With UV-B, egg hatch and the survival times of early and late instar nymphs declined in proportion to UV-B dose. A dose of 57.6 kJ m^−2^ d^−1^ reduced the survival time of only adult females. Female fecundity was reduced at high UV-A and UV-B doses but increased at low doses. The Bunsen–Roscoe law held true for eggs and early instar nymphs for different durations and irradiances of UV-B. Eggs and nymphs had ED_50_ values for UV-B lower than the daily fluxes of this wavelength experienced worldwide. Thus, UV-B could be a factor causing the psyllid to be scarce at high elevations.

## 1. Introduction

UV radiation is currently receiving much attention because of its deleterious and beneficial effects on living organisms [1,2,3,4,5]. The deleterious effects are primarily due to damage to DNA, which results in the production of pyrimidine dimers, and to the formation of reactive oxygen species (ROS) that generate oxidative stress [6,7,8,9,10]. The beneficial effects of exposure to UV radiation are caused by preferentially damaging less UV-tolerant predators [11,12], competitors [13,14], and pathogen [15]. Ultraviolet (UV) radiation emitted by the sun and artificial sources is typically divided into three wavebands: UV-C (200–280 nm), UV-B (280–315 nm), and UV-A (315–400 nm) [16]. This division is based on the absorbance of the waveband by atmospheric gasses and by their biological effect. UV-C is almost completely absorbed by ozone and other gases, UV-B is substantially absorbed, whilst UV-A is hardly absorbed; hence, approximately 95% of UV radiation reaching the Earth’s surface is UV-A. However, UV-B has more energy per photon and is more damaging to biological systems [1,17]. The ability to perceive UV radiation is common in vertebrates [18], and many insects possess trichromatic vision involving opsins that allow them to see wavelengths of 325–400 nm in the UV-A region, thereby allowing them to display many types of phototactic behaviour [19]. In addition to work on vision, many studies on the effects of UV radiation on insects have concentrated on the dose responses to UV-C, with the aim of using radiation of this waveband as a non-chemical means of controlling insects, particularly those that are storage pests [20,21,22]. Relatively few studies have examined the effects wavelengths of longer than those of UV-C.

*Diaphorina citri* Kuwayama (Hemiptera: Psyllidae), the Asian citrus psyllid (ACP), is the most serious pest of citrus in the world [23,24,25,26]. The psyllid is a relatively minor pest in the absence of the pathogen ‘*Candidatus* Liberibacter asiaticus’ (α-Proteobacteria) [24,27], the organism associated with the severe Asian form of huanglongbing and for which it is the major vector. The psyllid’s life cycle includes an egg stage, five nymphal instars, and the adult stage [28]. ACPs usually feed on abaxial sides of leaves [29], and they lay their eggs on the unopened new shoots of citrus plants [28]. The psyllid is able to perceive UV-A between 365 and 370 nm [19], and responses to UV radiation appear to involve both positive [30,31] and negative phototaxis [32,33].

This current study was stimulated by findings that populations of *D. citri* in Central Java, Indonesia, and Bhutan decline with elevation [34]. In Bhutan, the psyllid was found up to elevations of ~1200 m above sea level (ASL) but rarely higher, near Damphu, Tsirang (27.0073° N, 90.1242° E, 1571 m ASL) [35]. In Java, psyllid populations declined at elevations over 700 m ASL, most likely due to the effects of seasonal rainfall and fewer annual generations of the insect. Psyllid populations are known to be affected by a number of natural factors, including relative humidity, rainfall, ambient temperature, predators, parasitoids, and entomopathogens. However, a study by Om et al. [35] indicated that there was little evidence of these factors being responsible for the rarity of the psyllid at the higher altitudes in Bhutan, and it was hypothesised that UV radiation may be responsible. Although work has been performed on the ability of the psyllid to perceive UV radiation, no studies have been made on the doses of the different wavebands of UV radiation on its development or survival. However, it has been shown that numbers of adults reared in cages were positively correlated with the duration of illumination and with the irradiation of light with wavelengths between 400 and 1100 nm [36]. Therefore, this study was undertaken to determine the doses of UV-A and UV-B radiation that affect eggs, early and late instar nymphs, and male and female adults of *D. citri*. The dose of UV radiation received by an organism is a function of its irradiance and duration. For many photochemical reactions, the effects are proportional to dose, thus obeying the Bunsen–Roscoe reciprocity law [37,38]. However, for different time × irradiance combinations, the effects may not be reciprocal and can be heavily influenced by the power of the irradiance [38]. Therefore, compliance with the reciprocity law for different treatment combinations was also examined.

## 2. Materials and Methods

### 2.1. Materials

#### 2.1.1. Insect Culture

The adult *D. citri* used in this study were taken from a long-term (3-year-old) culture maintained by the Laboratory of Insect Ecology of the South China Agricultural University on healthy *Murraya paniculata* (L.) Jack at 25–28 °C. The culture was established from psyllids present on the campus at the time. Fluorescent light was used, both in the laboratory and in the incubator used to rear the insect. The light was kept at 1.5 m height from the floor where potted *M. paniculata* were placed along with the ACP. The lights were turned on in the evenings, as the laboratory did not need lights during the day. The photoperiod was set to a 12:12 h L:D cycle and the relative humidity to 60–65%. Adults to be used for experimentation were collected and exposed to UV radiation during the mornings.

#### 2.1.2. Ultraviolet Light

UV-A radiation was produced by a 30 W LED lamp (model KEM-HP030FL-365), and UV-B radiation was also generated by a 30 W LED lamp (model KEM-HP030UVL) supplied by the Guangzhou Langke Photoelectric Co., Ltd., Guangzhou, China. The lamps were supported by a bracket (L × D × H: 1500 × 4500 × 550 mm), and a black curtain was used to cover the experimental structure during exposures. After exposure, treated psyllids were placed in a climate-controlled box under constant white fluorescent light. To calibrate the UV irradiance, a UV-A meter (320–400 nm; λP = 365 nm) and UV-B meter (λ: 275–330 nm; λP = 297 nm) were purchased from the Shanghai Gaozhi Precision Instrument Co., Ltd. (Shanghai, China).

### 2.2. Psyllid Preparation and Observation after Irradiation with Either UV-A or UV-B

#### 2.2.1. Effects on Egg Hatchability

Fifteen mature adults were collected and introduced onto new shoots of *M. paniculata* that were kept in Murashige and Skoog nutrient solution [39]. After 24 h, the adults were removed with the help of an aspirator, and the eggs laid on the shoots were counted under a stereomicroscope (SFC-11 series, Motic Microscopes, Germany). In a preliminary experiment, it was found that most of the eggs hatched on the 5th day after being laid. The eggs were exposed immediately to UV-A or UV-B for different periods at different irradiances. Control eggs were prepared similarly but were not exposed to UV radiation. For UV-A, four experiments were conducted, each consisting of six treatments. The eggs were exposed daily to 8, 12, 25, and 107 W m^−2^ for 0, 30, 60, 90, 120, and 180 min. In the final experiment, eggs were also exposed to UV-A at 107 W m^−2^ for 240 and 300 min each day to give eight treatments. For UV-B, three experiments were conducted, each with six treatments. The eggs were exposed daily to 1.4, 1.5, and 1.7 W m^−2^ for 0, 30, 60, 90, 120, and 180 min. In the final experiment using this waveband, eggs were exposed to 1.7 W m^−2^ for 240 and 300 min each day. Eggs were exposed to UV-A or UV-B light on each of four consecutive days. Each experiment was independently replicated four times on separate occasions with 24–185 eggs per treatment for UV-A and 20–100 eggs for UV-B being used per treatment per occasion. The nymphs were observed every day after egg hatching until they became adults or died and were transferred to new tender shoots of *M. paniculata* after the third day.

#### 2.2.2. Effects on Early Instar Nymphs

Ten mature adults were placed on new shoots of *M. paniculata*. After 24 h, the adults were removed with the help of an aspirator, and the eggs laid on the new shoots were counted with the help of a stereomicroscope. On the second day after the eggs started to hatch, the numbers of nymphs and unhatched eggs were counted and the nymphs exposed to UV-A or UV-B. The shoots with nymphs were irradiated for 5 days for respective doses of UV-A or UV-B. For UV-A, five experiments were conducted, each consisting of four treatments. The nymphs were exposed daily at 39, 49, 59, 69, and 77 W m^−2^ for 0, 60, 12, and 180 min. For UV-B light, six experiments were conducted, each with four treatments. The nymphs (20–100 nymphs per replication) were exposed daily at 0.45, 0.6, 0.81, 1, 1.3, and 1.6 W m^−2^ for 60, 120, and 180 min. Nymphs acting as controls were treated as above but were not exposed to UV radiation. Each experiment was independently replicated on four separate occasions with 20–100 nymphs per treatment per occasion being used for both UV-A and UV-B. 

#### 2.2.3. Effects on Late Instar Nymph

Nymphs were prepared as detailed above. They were exposed to UV-B on the 9th day after egg hatching, and the shoots with nymphs were irradiated for 5 days. This work consisted of three experiments with four treatments per experiment. The nymphs were exposed daily at 0.6, 0.81, and 1 W m^−2^ for 0, 60, 120, and 180 min. Twenty-five nymphs were used in each replication, and each treatment was replicated four times. 

#### 2.2.4. Effects on Adult Longevity and Reproduction

Male and female adult psyllids were identified based on their morphology, and 30 pairs of three-day-old adults were exposed to either UV-A or UV-B. The pairs were confined in small (30–50 mm diameter) Petri dishes covered with a UV-transparent plastic film containing a small ventilation hole. Each Petri dish contained a mature citrus leaf partially embedded in agar. Each pair of psyllids was then reared in a plastic cup (150 mm in height); each pair was treated as one replicate. The open end of the plastic cup was closed with a net cover of degreased cotton yarn. Adults were exposed daily to UV-A for 0 (control), 1, 4, and 7 h per day at 35 W m^−2^, and for UV-B, adults were exposed at 0 (control), 1, 4, 7, 9, and 16 h per day at 1 W m^−2^ each day until all of the males and females died. In a separate experiment, adult females were exposed daily to 0 and 57.6 kJ m^−2^. The temperature in the irradiated area was 25–28 °C. After irradiation, the adults of each treatment were kept in the controlled environment chamber at 25–27 °C and 60–65% relative humidity. 

The effect of UV radiation on the following reproductive parameters of the adult females was recorded: the pre-oviposition period (the duration before oviposition of the first eggs), the oviposition period (the duration that females laid eggs), and the post-oviposition (the duration after the cessation of oviposition and the death of the female). The number of eggs laid per female per day throughout the whole oviposition period and the longevity of males and females were recorded daily. The number of eggs was counted using a stereomicroscope. The eggs laid by the female of the P generation were referred to as the F_1_ generation. The number of newly hatched nymphs was recorded daily, and the number of unhatched eggs was recorded at the end of the experiment. The number of adults and timing of eclosion were also recorded. Ten pairs of adults from each treatment were randomly selected to study the carryover effects of UV radiation on the development and longevity of the F_1_ generation. 

### 2.3. The Applicability of the Bunsen–Roscoe Reciprocity Law to Immature Stages of D. citri

#### 2.3.1. Eggs

Psyllid eggs and new shoots of *M. paniculata* were prepared as described above. After being laid, the eggs were immediately exposed to UV-B. This section of the study consisted of a single experiment with five treatments with the following combinations of irradiances and exposure times: 0 and 0.71 W m^−2^ for 317 min (13.50 kJ m^−2^); 0 and 0.37 W m^−2^ for 609 min (13.52 kJ m^−2^); 0 and 0.6 W m^−2^ for 375 min (13.50 kJ m^−2^); 0 and 0.51 W m^−2^ for 441 min (13.49 kJ m^−2^); and 0 and 0.31 W m^−2^ for 726 min (13.50 kJ m^−2^). The doses of UV-B were designed to be approximately equal to the ED_50_ for eggs and were given daily. Each combination of dose was replicated on four separate occasions, with the numbers of eggs used for each combination of UV-B intensity and irradiation time being 21–136.

#### 2.3.2. Early Instar Nymphs

Early instar nymphs of *D. citri* were prepared as described above. On the 4th and 5th days, the status of the eggs was checked. The nymphs were counted, and the unhatched eggs were removed. The shoots with nymphs were exposed to the following combinations of irradiances and exposure times of UV-B, designed to be approximately equal to the ED_50_ for nymphs: 0 and 1.1 W m^−2^ for 60 min (3.96 kJ m^−2^ d^−1^); 0 and 0.75 W m^−2^ for 90 min (4.05 kJ m^−2^ d^−1^); 0 and 0.56 W m^−2^ for 120 min (4.03 kJ m^−2^ d^−1^); 0 and 0.37 W m^−2^ for 180 min (3.99 kJ m^−2^ d^−1^); and 0 and 0.28 W m^−2^ for 240 min (4.03 kJ m^−2^ d^−1^) for 5 days. Each treatment was repeated four times, with 20–70 nymphs used per replication. After exposure to UV-B, the mortality rates from the different intensity treatments were compared. 

### 2.4. Data Analysis

To determine differences among treatment means, analyses of variance (ANOVA) and Duncan’s multiple range tests at *p* = 0.05 were performed. The data pertaining to egg hatch and nymphal survival were arcsine-transformed before analysis. Other data sets were checked for heteroscedasticity prior to analysis using Bartlett’s test and, where heteroscedasticity was found, the data were appropriately transformed. Regression of mortality on cumulative UV irradiance and the ED_50s_ were calculated using probit analysis. The mortality of eggs or nymphs in each treatment was corrected using Abbott’s formula [40]. The corrected mortalities were transformed to probits, with data excluded for mortalities of 0 or 100%. Kaplan-Meier survival analysis was used to determine the effect of UV on the survival of nymphs. Linear regression analysis was applied to the data to relate the dose with egg hatching and nymph mortality. 

To evaluate whether doses of UV radiation given with different irradiance and time combinations showed reciprocity, the mortality data were converted to probits, and probit mortality was regressed against dose. The data were also statistically analysed by ANOVA, and the means were compared using a Duncan’s multiple range test. 

Statistical analyses were performed using Microsoft Office Excel 2019 or SPSS 25.0.

## 3. Results

### 3.1. Effects of UV-A on Egg Hatch and Survival of Emerging Nymphs

The results of the data pertaining to UV-A showed that the hatching of *D. citri* eggs was affected by UV-A, but the effect was not marked. Exposure to higher doses of UV-A reduced egg hatching of *D. citri*, and the effects gradually increased with increasing dose (Appendix A). However, even at the highest dose (1926 kJ m^−2^ d^−1^), the reduction in hatching was only ~22%. As a consequence, when the data for all experiments were combined, the relationship between the UV-A dose and percentage hatch was weak (Figure 1), and the relationship between dose and probit-transformed hatch was not significant (Table 1). The ED_50_ value for eggs was estimated to be 4.21 × 109 kJ m^−2^ d^−1^. The mean survival times of the emerging nymphs (Appendix A) reduced only at the higher radiation doses used in Expts 3 and 4; however, within these experiments, the data were variable and the relationship with dose was weak.

### 3.2. Effects of UV-B on Egg Hatch and Survival of Emerging Nymphs

As expected, in contrast to UV-A, the hatching of *D. citri* eggs was strongly affected by exposure to UV-B. Daily doses of UV-B above 2.51 kJ m^−2^ reduced hatching compared to controls, and the effects gradually increased with increasing dose (Appendix A). Irradiation with UV-B with a dose of 30.6 kJ m^−2^ reduced egg hatching to nearly 20% of that of the controls. When the percentage of egg hatching was regressed against the UV-B dose, a strong, negative, linear relationship was obtained (Figure 2). The ED_50_ was estimated to be 13.32 kJ m^−2^ d^−1^ (Table 2). Kaplan–Meier analysis showed that the survival of nymphs hatching from eggs irradiated with different doses of UV-B was strongly affected by the higher doses applied to the eggs (Appendix A).

### 3.3. Effects of UV-A and UV-B on Early Instar Nymphs and UV-B on Late Instars

The survival of early instar nymphs of *D. citri* was little affected by UV-A (Appendix A) and, when survival was plotted against the dose, a weak, linear, negative relationship was obtained (Figure 3). Adults were only obtained in Expt 5, and only from the 77 W m^−2^ (277.2–831.6 kJ m^−2^ d^−1^) group. The ED_50_ estimated from the combined data was 324.63 kJ m^−2^ d^−1^. The proportion of adults emerging from the treated and untreated nymphs was low; however, there was a significant (*F* = 6.337, *p* = 0.003) increase in the number of days adults took to develop when nymphs were treated at greater than 550 kJ m^−2^ each day.

As with the eggs, the survival times of early instar nymphs were affected by UV-B (Appendix A), and there was a negative, linear relationship between survival and the dose (Figure 4). The ED_50_ (4.01 kJ m^−2^ d^−1^) was three times lower for the nymphs as compared to eggs when irradiated with UV-B (Table 2). Adults only emerged from the nymphs in Expt 1 treated at 0.45 and 0.6 W m^−2^ for 1 and 2 h (1.62–4.86 kJ m^−2^ d^−1^). Although the proportion of adults emerging from either treated or untreated nymphs was again low, there was a significant (*F* = 30.682, *p* < 0.0001) increase in the number of days individuals took to develop from nymphs to adults after irradiation. 

The mortality of late instar nymphs exposed to UV-B radiation increased with an increasing dose of UV-B (Figure 5). The effects of UV-B irradiation on late instar nymphs were less than that experienced for early instars; the ED_50_ for the late instars was estimated to be 6.17 kJ m^−2^ d^−1^.

### 3.4. Effects of UV-A and UV-B Radiation on Adult Longevity and Survival

The longevity of adult male and female *D. citri* was reduced (*F* = 4.529, *p* = 0.05 for males; *F* = 7.185, *p* < 0.0001 for females) after exposure to UV-A at doses of 504 kJ m^−2^ d^−1^ (Appendix A) or greater. For UV-B, there was no effect of UV-B at doses up to 32.4 kJ m^−2^ d^−1^. However, a UV-B dose of 57.6 kJ m^−2^ d^−1^ significantly (*p* < 0.0001) reduced the survival of adult females but not the survival of males (*p* = 0.1888) (Figure 6). 

### 3.5. Effects of UV-A Radiation on Reproduction

The pre-oviposition, oviposition, and post-oviposition periods of the adult females were not different from those of the control groups after irradiation with various doses of UV-A (Table 3). Moreover, the fecundity (mean total number of eggs laid per female throughout their whole oviposition period) was only marginally affected by exposure to UV-A (Appendix A); however, the pattern of egg production was similar to that seen in the UV-B data (see below). Female *D. citri* exposed to 126 kJ m^−2^ laid more eggs than those in the control, whereas those exposed to 504 and 882 kJ m^−2^ laid fewer; however, the effect was not statistically significant (*F* = 2.13, *P* = 0.1054). Among the 30 females in each treatment, 22 untreated females, 20 females treated at 126 kJ m^−2^, and 18 females at 504 kJ m^−2^ laid eggs, whereas only 6 females laid eggs when treated at 882 kJ m^−2^.

### 3.6. Effects of UV-B Radiation on Reproduction

The pre-oviposition periods of females irradiated with UV-B were not significantly affected when adults were treated with 3.6–25.3 kJ m^−2^ d^−1^ (Table 4). The oviposition period was slightly longer when adults were treated at 3.6 kJ m^−2^ d^−1^ but shorter at 25.2 kJ m^−2^ d^−1^. However, the post-oviposition period significantly increased at 3.6 kJ m^−2^ compared to the control group. When adults were treated at 57.6 kJ m^−2^, the pre-oviposition (*p* = 0.024) and oviposition (*p* = 0.035) period significantly decreased (Table 5) and the post-oviposition period slightly increased but did not significantly differ from the control (*p =* 0.58).

Exposure to UV-B significantly affected oviposition (*F* = 6.096, *p =* 0.001). There was a trend for exposure to UV-B at 3.6 kJ m^−2^ d^−1^ (1 W m^−2^ for 1 h) to increase fecundity above that recorded in the control (Appendix A); however, the difference was not statistically significant. Exposure to the higher doses tended to decrease oviposition compared to control. Females in the 3.6, 14.4, 25.2, and 57.6 kJ m^−2^ d^−1^ treatments laid an average of 129.3, 88.7, 43.05, and 24.14 eggs per females, respectively. In a separate experiment, females treated at 57.6 kJ m^−2^ d^−1^ laid fewer eggs compared to the control group, but the difference was not statistically significant (*p =* 0.106) (Appendix A). Among the 30 females in each treatment, 22 untreated females, 19 females treated at 3.6 kJ m^−2^ d^−1^, 27 females at 14.4 kJ m^−2^ d^−1^, 21 females treated at 25.2 kJ m^−2^ d^−1^, and 7 females treated at 57.6 kJ m^−2^ d^−1^ laid eggs; no females laid eggs when treated at 32.4 kJ m^−2^ d^−1^.

### 3.7. Effect of UV-A and UV-B Radiation on the Development of the F_1_ Generations

UV-A significantly affected the hatching of eggs of the F_1_ generation when adult females were exposed at 882 kJ m^−2^ d^−1^ (*F* = 11.29, *p* < 0.0001) (Appendix A). However, survival of the F_1_ nymphs was not significantly affected by exposure to this waveband (*F* = 1.292, *p* = 0.291). The 882 kJ m^−2^ d^−1^ data were excluded from this analysis because only six females (one in each of six pairs of adults) laid eggs, most of which did not hatch. There was no difference in the time taken by individuals to develop from nymphs to adults (male: *F* = 0.33, *p* = 0.726; female: *F* = 3.049, *p* = 0.085) (Appendix A). 

At the doses given, treatment with UV-B did not significantly affect the percentage of eggs hatching, nymphal longevity, nor the times to eclosion of adult males and females (Appendix A). 

### 3.8. Reciprocity Law for Eggs and Early Instar Nymphs

Eggs and early instar nymphs were exposed to five different combinations of irradiances and durations of UV-B, giving doses designed to approximate their relevant ED_50s_. Both sets of data were subjected to ANOVA; there were no significant differences between the treatments in either data set (eggs: *F* = 1.773, *P* = 0.187; nymphs: *F* = 0.893, *P* = 0.492). Both data sets were also subjected to regression analysis (Figure 7 and Figure 8), regressing irradiance against probit mortality; the resulting coefficients of determination were low for both stadia (eggs: R^2^ = 0.203, *p* = 0.186; nymphs R^2^ = 0.051, *p* = 0.876).

## 4. Discussion

The effects of UV on various arthropods have been studied previously, and all have shown that UV-A and, particularly, UV-B can be harmful and affect their biology and physiology [41,42,43,44,45]. Our laboratory-based study indicates that UV-A has little effect on the immature stages of *D. citri*; UV-B, however, was detrimental. There was a significant reduction in hatching when eggs were irradiated with doses of UV-B > 5.04 kJ m^−2^ d^−1^. However, UV-A only reduced egg hatch when the applied doses were >1500 kJ m^−2^ d^−1^, and at the highest dose used (1926 kJ m^−2^ d^−1^) nearly 70% of eggs hatched, indicating that UV-A had little to no effect at the doses applied. Although studies have been made of UV radiation on arthropods, most have been on the effects of UV-C. Few have studied the effects of UV radiation on eggs, and only the study of Murata and Osakabe [46] has determined an ED_50_ for this stadium (0.58 kJ m^−2^ for two-spotted mite, *Tetranychus urticae* Koch (Trombidiformes: Tetranychidae)) exposed to predominantly UV-B. This study also showed that eggs of *T. urticae* were the most sensitive stadium of the four tested (eggs, larvae, teleiochrysalises, and adults). A study by Alwaneen et al. [47] with *Cadra cautella* Walker (Lepidoptera: Pyralidae) using unspecified doses of UV-B (280–315 nm) also showed that eggs were the most sensitive of the stadia tested (eggs, larvae, and adults). However, in our study, nymphs were more sensitive than eggs (see below). Guerra et al. [48] found that the eggs of *Heliothis virescens* F. and *H*. *zea* (Boddie) (Lepidoptera: Noctuidae) were little affected by various unspecified doses of UV-A (360 nm).

Damage to eggs resulted in them shrinking when they were exposed to high doses of UV-B; however, the control eggs and eggs treated with UV-A remained in good condition. Eggs of the tropical warehouse moth, *Cadra cautella*, were “shrunken and appeared damaged” when exposed to UV-B [47]. Similarly, eggs of the *Smittia* spp. (Diptera: Chironomidae) were affected when exposed to UV-B (280 and 300 nm), but were not affected by treatment at wavelengths > 310 nm [49]. Wang et al. [50] reported that the effect of UV radiation was reduced when zebrafish *Danio rerio* (F. Hamilton) (Cypriniformes: Cyprinidae) embryos were fully enclosed by an artificial mineral-shell. This suggests that if eggs are robust or have protective coverings they can be protected from the harmful effect of UV radiation. Freshly deposited eggs of *D. citri* are light yellow, turning bright orange at maturity [28]. Some studies suggest that egg coloration plays an important role in protecting them from the harmful effects of solar UV radiation. Female spined soldier bugs (*Podisus maculiventris* (Say) (Hemiptera: Pentatomidae)) were found to lay darker eggs on the upper surface of leaves to protect them from the negative effect of UV exposure [51]; the pigmentation of the darker coloured eggs of this species also gave protection from the adverse effects of UV-A [52]. Yellow pigmentation has also been shown to protect adult springtails (Collembola) [53] and termites (Blattodea: Isoptera) [54] from the effect of UV radiation; the pigment in the eggs of *D. citri* may serve a similar purpose.

At the doses applied, irradiation with UV-A had little negative effect on the survival of early instar nymphs. With respect to UV-B, nymphs at this development stage were more susceptible than eggs, with late instar nymphs showing a greater tolerance to this waveband than early instar nymphs but less than that of the eggs. The soft transparent body of early instars may allow greater penetration of UV-B and more subsequent damage: alternatively, their physiology may be disrupted. Early instars survived for longer periods when exposed to UV-B at less than 4.68 kJ m^−2^ d^−1^, and some developed into adults. However, as the UV-B dose increased, development quickly stopped. For late instar, doses greater than 6 kJ m^−2^ d^−1^ of UV-B had negative impacts on survival. The early instar nymphs of *Periplaneta americana* (L.) (Blattodae: Blattidae) and *Blatella germanica* (Blattodae: Ectobiidae) were also found to be sensitive to single doses of UV radiation, with wavelengths less than 297 nm at 0.33 kJ m^−2^ [41], and Ali et al. [55] showed that the early instars of the noctuid *Mythimna separata* Walker (Lepidoptera: Noctuidae) were also more susceptible than later instars to daily doses of UV-A. The nymphs of *D. citri* become more pigmented as they pass though their larval stages [28], which would afford them greater protection against UV radiation. In addition, protein and fat levels are lower in earlier instars and higher in later instar nymphs [55]; these molecules have a role in immune defence and may also help mitigate the effects of UV radiation.

Adults of both sexes were the most tolerant stadium to UV radiation. The longevity of both adult males and females only decreased significantly when exposed to greater than 504 kJ m^−2^ d^−1^ of UV-A. This value is higher than was found for the cotton bollworm (*Helicoverpa armigera* (Hübner) (Lepidoptera: Noctuidae)), where daily doses of UV-A (320–400 nm) of 10.8 kJ m^−2^ or greater reduced longevity [8], and for the citrus whitefly (*Dialeurodes citri* (Ashmead) (Hemiptera: Aleyrodidae)), where a dose of 11.52 kJ m^−2^ d^−1^ (320–400 nm) or greater also affected longevity [56]. These differences may be due to innate differences between the species or to differences in experimental procedures, including those that affect photoreactivation (see below). In addition, adult *D. citri* are darkly pigmented, and the pigment may give protection against UV radiation. The dark pigment of the collembolan *Orchesella cincta* (L.) (Collembola: Entomobryidae) absorbs in the UV-A waveband [57]. 

There was little difference between the susceptibility to UV-A of the sexes of *D. citri*. The effect of UV-B was more severe, with female longevity reduced by a dose of 57.6 kJ m^−2^ d^−1^; however, the longevity of males was not affected by this dose. Differences between the sexes have been found by Ali et al. [55], where the longevity of adult females of *Mythimna separata* were more sensitive to UV-A than males, and by Okamoto [58], who showed that males of *B. germanica* are more susceptible than females to UV-C. Further work is required to define better the doses of this waveband that are deleterious to both sexes of *D. citri* and to determine why the differences in susceptibility occur.

Although the treatments with UV-A or UV-B did not affect the oviposition period, the fecundity of *D. citri* was increased at the lowest doses of UV-A and UV-B, before declining at higher doses. This stimulatory effect of low doses of radiation appears to be common and has been found in the growth and reproductive rates of *Sitobium avenae* F. (Hemiptera: Aphididae) treated with UV-B [45]. The phenomenon has also been found for other wavebands: for the oviposition of *Dialeurodes citri* [56], *Mythimna separata* [55], and *Helicoverpa armigera* [8] treated with UV-A; for the longevity of *Drosophila melanogaster* Meigen and *Drosophila subobscura* Collin (Diptera: Drosophilidae) treated with X-rays [59,60,61]; and for the longevity of *Musca domestica* L. (Diptera: Muscidae) treated with γ-rays [62]. A first explanation given for this hormetic response is the induction of stable epigenetic modifications to DNA and an enhanced DNA repair capacity, as shown by increased resistance to DNA cleavage with a single-strand-specific S1 nuclease [61,63,64]. A second explanation is that the low irradiation treatments increase the synthesis of vitamin D, with the resulting benefits this brings. Oonincx et al. [65] have shown that insects given low doses of UV-B are able to synthesis vitamin D. In this study, there was little effect of either UV-A or UV-B on the F_1_ generations produced from irradiated parents. This differs from similar studies on *Dialeurodes citri* [56], *Mythimna separata* [55], and *Helicoverpa armigera* [8], all of which found reduced survival and prolonged development times of F_1_ generations. Further work is needed to clarify the effects of UV radiation on the offspring of irradiated *D. citri*.

We also observed that the Bunsen–Roscoe reciprocity law applies to *D. citri* eggs and early instar nymphs treated with UV-B. Egg hatching and survival of early instar nymphs were negatively correlated with cumulative UV-B exposure, independent of UV-B intensity. The validity of the Bunsen–Roscoe law on the biology of insects has not been investigated. However, two studies have explored the relationships between dose and irradiance using *Tetranychus urticae* subjected to various treatments with UV-B. Both studies [46,66] found that the hatchability of eggs; the mortality of eggs, larvae, teleiochrysalises, and adult females; and the development time from larvae to adult emergence were all linearly related to dose, regardless of intensity. Hence, under the range of doses and irradiance times used, all three studies have shown the reciprocity law was upheld. The applicability of the law needs to be examined in a greater range of species because it does not apply to the development and survival of amphibians [67] and aquatic invertebrates [68]. 

To determine how the doses of UV-A and UV-B determined to have affected the psyllid’s development and survival relate to those occurring naturally, we compared the derived ED_50s_ with daily doses from various locations around the world. Unfortunately, data from Bhutan were not available. On a cloudless summer day in Melbourne (37.8130° S, 144.9631° E, 22 m ASL), Australia, total UV radiation doses were found to be 1467 kJ m^−2^, 95% of which was due to UV-A (1387 kJ m^−2^) and 5% (80 kJ m^−2^) to UV-B. Equivalent figures for a winter’s day were 452 kJ m^−2^, 439 kJ m^−2^ (97%), and 14 kJ m^−2^ (3%) (data supplied by the Australian Bureau of Meteorology). In Valladolid (41.6532° N, 4.7246° E, 692 m ASL), Spain, during 2002–2011, the maximum daily UV-B dose in May and June was 50.24 kJ m^−2^ and the minimum was 0.88 kJ m^−2^ in January and December [69]. Escobedo-Bretado et al. [70] reported the mean monthly accumulated UV-A dose at Durango, Mexico in December 2015 to be 905 kJ m^−2^·day^−1^ and in June 2016 to be 1732 kJ m^−2^·day. The maximum energy values occurred in July with 2077 kJ m^−2^·d^−1^ for UV-A and 60 kJ m^−2^·d^−1^ for UV-B: the minimums were found in November and were 255 kJ m^−2^·day^−1^ for UV-A and 6.5 kJ m^−2^·day^−1^ for UV-B. The daily doses of UV-B reported from Australia, Spain, and Mexico are all higher than the ED_50s_ for UV-B for *D. citri* found in this study. The effects of these exposures to radiation would be mitigated due to photoreactivation [71], the partial reversal of effects of far-UV by subsequent treatment with near-UV or visible radiation [72,73]. The effects of the UV radiation experienced by *D. citri* in this study would also be mitigated by the same phenomenon. The effects of solar radiation would also be reduced by *D. citri*’s preference for the underside of leaves, which would reduce its exposure to direct radiation but still leave it exposed to potentially detrimental levels of diffuse radiation. Diffuse radiation would be especially high and shading low in the outer regions of canopies where the psyllid prefers to reside and where new growth and light intensities and durations are favourable for oviposition [74,75,76]. Percentages of UV-B and UV-A in diffuse light within shaded evergreen canopies measured at Toowoomba (27.61° S, 696 m ASL), Queensland, ranged from 23 to 59% and 17 to 31%, respectively, of global UV radiation [77]. Therefore, the effects of shading may not be as great as would expected, and this study suggests that UV radiation may be a factor affecting the distribution of *D. citri*.

Solar UV radiation increases with altitude. Sullivan et al. [78] have shown that UV-B increased from 10.8 kJ m^−2^ at sea level to 12.4 kJ m^−2^ at 3000 m ASL across an elevational gradient in Hawaii; other studies show an increase of 10–30% per 1000 m depending on the wavelengths of the UV radiation [79,80]. *D. citri* populations decline with elevation. In addition the studies in Bhutan and Indonesia [28] outlined in the Introduction, the insect was found up to ~1500 m ASL but was absent above ~1700 m ASL in Saudi Arabia [81]. Similarly, it has been recorded up to 1200 m ASL at localities between 26.00 and 29.00° N in China. In Puerto Rico, *D. citri* was not found above 600 m ASL [82]. Along with the pest, the incidence of huanglongbing has also been found to decline with altitude [35,82,83]. The Himalayan region is characterised as an area that receives high amounts of UV radiation [84,85,86,87,88,89], and the absence of *D. citri* at the higher altitudes found in the study by Om et al. [35] may be due to amounts passing a critical dose sufficient to either directly affect the insect or through indirect effects on its host plants or microbiome. ‘*Candidatus* Carsonella ruddii’, the primary endosymbiont of psyllids [90,91], lacks the ability to repair damaged DNA [92], which may render it susceptible to UV radiation. In conclusion, irrespective of whether UV radiation is a factor influencing psyllid populations, the studies outlined above showing the decline in psyllid populations with altitude open up the possibility of producing citrus at higher altitudes with minimal threat from huanglongbing and for locating nursery production facilities at such altitudes so that disease-free propagation material can be produced.

## Figures and Tables

**Figure 1 insects-14-00445-f001:**
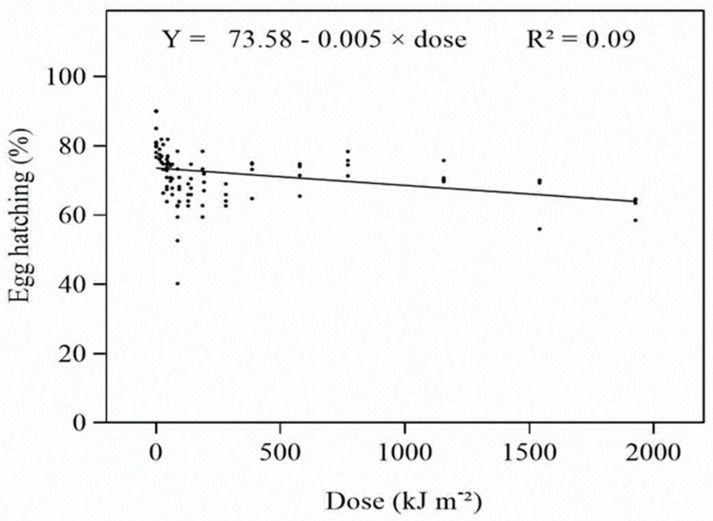
Relationship between daily UV-A dose and percentage egg hatch.

**Figure 2 insects-14-00445-f002:**
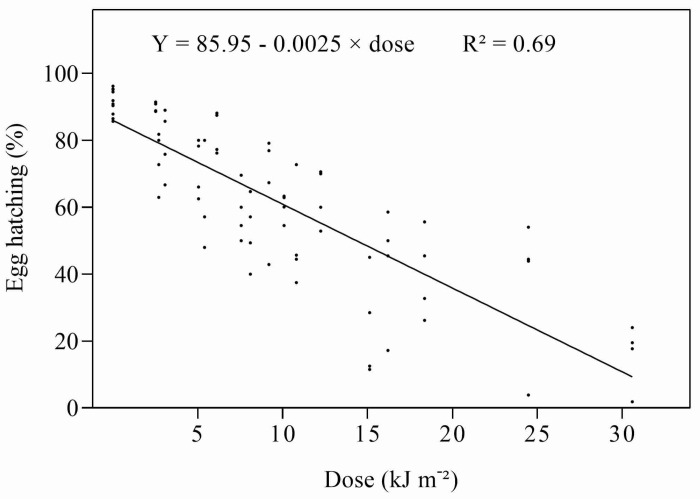
Relationship between daily UV-B dose and percentage of egg hatch.

**Figure 3 insects-14-00445-f003:**
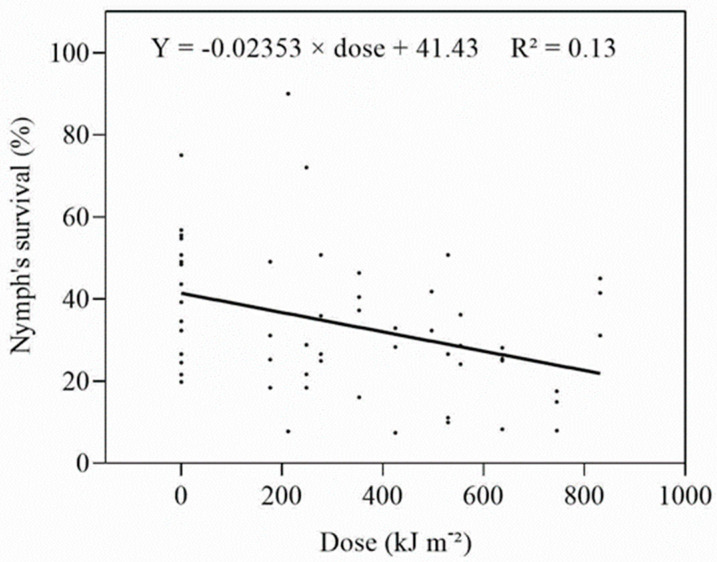
Relationship between daily dose and survival of early instar nymphs irradiated with UV-A.

**Figure 4 insects-14-00445-f004:**
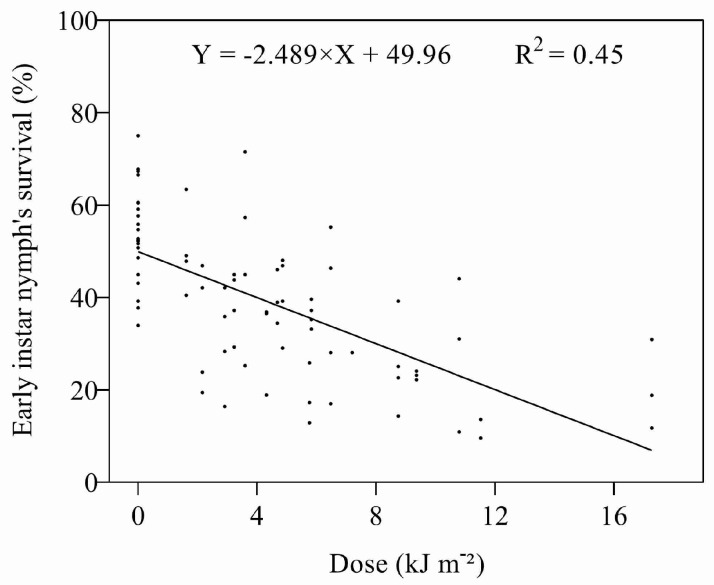
Relationship between UV-B dose and survival of early instar nymphs.

**Figure 5 insects-14-00445-f005:**
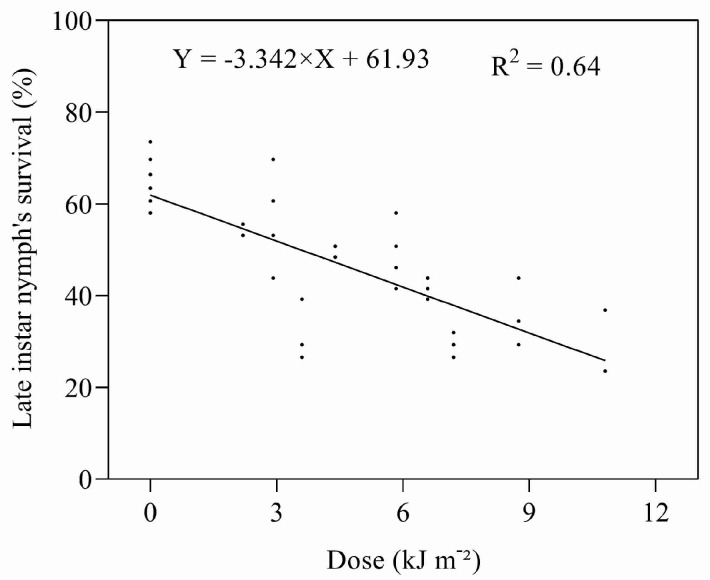
Relationship between UV-B dose and survival of late instar nymphs.

**Figure 6 insects-14-00445-f006:**
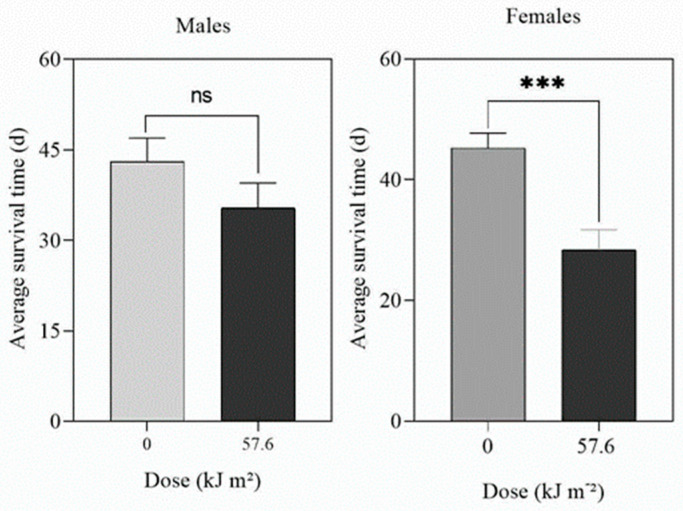
Effect of UV-B on the longevity of adult male and female *D. citri* treated for 16 h d^−1^ at 1 W m^−2^ until they all died. The data are means (*n* = 30) and the bars represent the standard errors. ns = not significant. *** = significant at *p* = 0.01.

**Figure 7 insects-14-00445-f007:**
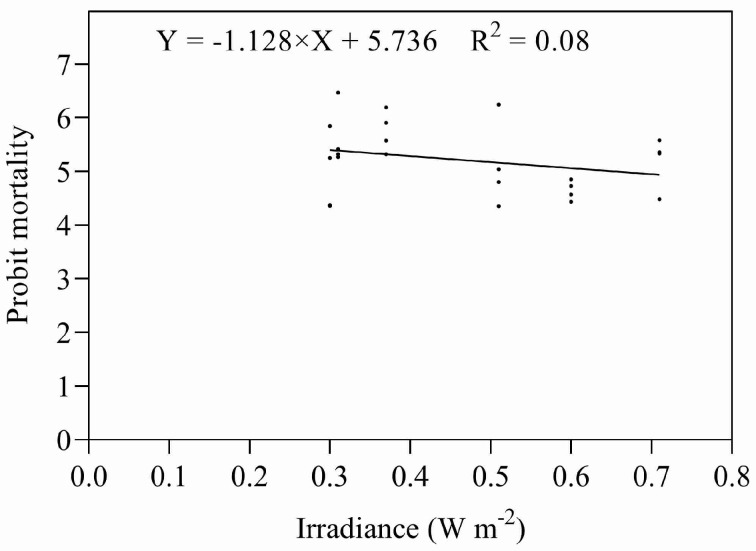
Mortality of eggs exposed to UV-B radiation at different combinations of intensity and duration.

**Figure 8 insects-14-00445-f008:**
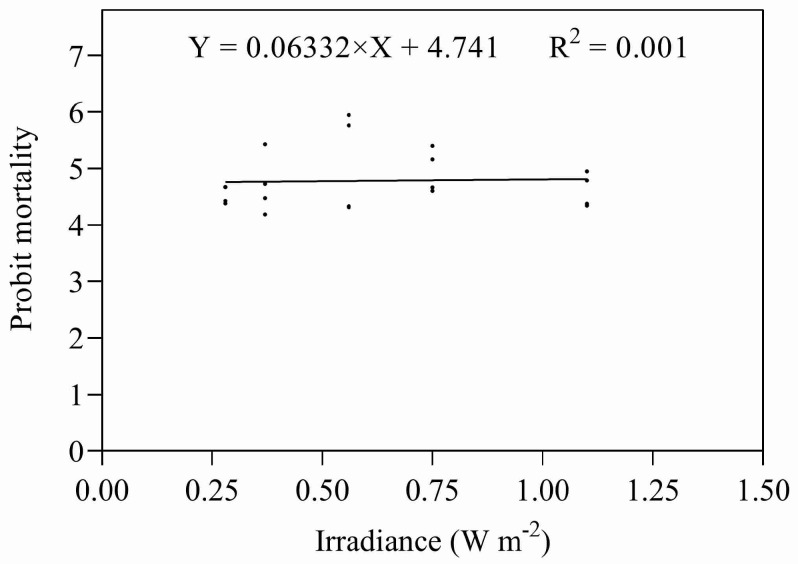
Mortality of nymphs exposed to UV-B radiation at different combinations of intensity and duration.

**Table 1 insects-14-00445-t001:** Linear regression of probit mortality on log10 UV-A dose (kJ m^−2^ d^−1^) and ED_50_ values of eggs and early instar nymphs of *D. citri*.

Developmental Stage	Regression Formula	R²	*p*	ED_50_	95% Confidence Interval of Gradient
				(kJ m^−2^ d^−1^)	
Eggs	Y = 0.1871x + 2.638	0.048	0.04	4.21 × 10^9^	0.007709 to 0.3666
Nymphs	Y = 0.8168x + 0.4983	0.045	0.19	324.63	−0.4126 to 2.046

**Table 2 insects-14-00445-t002:** Linear regression of probit mortality on log10 UV-B dose and ED_50_ values for eggs and nymphs of *D. citri.*

Developmental Stage	Regression Formula	R^2^	*p*	ED_50_ (kJ m^−2^ d^−1^)	95% Confidence Interval of Gradient
Eggs	Y = 1.944x − 3.018	0.6109	<0.0001	13.32	1.563 to 2.325
Early instar nymphs	Y = 1.695x − 1.108	0.2754	<0.0001	4.01	0.9224 to 2.469
Late instar nymphs	Y = 1.767x − 1.697	0.3649	0.0001	6.17	0.9413 to 2.592

**Table 3 insects-14-00445-t003:** Effect of UV-A on the reproductive parameters of adult females. The data are means (*n* = 6–22), and the bars represent the standard errors. Means annotated with different letters are significantly different from each other according to Duncan’s MRT tests at *p* = 0.05.

Dose (kJ m^−2^ d^−1^)	Pre-Oviposition Period (Days)	Oviposition Period (Days)	Post-Oviposition Period (Days)
0	14.56 ± 1.004 a	9.47 ± 0.99 a	4.69 ± 0.76 a
126	13.25 ± 1.14 a	10.45 ± 1.88 a	5.75 ± 1.58 a
504	12.38 ± 0.37 a	7.53 ± 0.78 a	4.47 ± 1.19 a
882	12.37 ± 0.46 a	8.50 ± 2.66 a	8.00 ± 3.098 a

Means annotated with different letters are significantly different from each other according to Duncan’s MRT tests at *p* = 0.05.

**Table 4 insects-14-00445-t004:** Effect of irradiation with UV-B on the reproductive parameters of adult females.

Doses (kJ m^−2^ d^−1^)	Pre-Oviposition Period (Days)	Oviposition Period (Days)	Post-Oviposition Period (Days)
0.0	14.56 ± 1.00 a	9.48 ± 0.99 ab	4.70 ± 0.76 b
3.6	13.78 ± 1.25 a	11.83 ± 1.77 a	7.83 ± 1.58 a
14.4	12.96 ± 0.55 a	10.46 ± 1.21 ab	3.30 ± 0.63 b
25.2	13.10 ± 1.078 a	7.09 ± 1.10 b	5.10 ± 1.24 ab

Means annotated with different letters are significantly different from each other according to Duncan’s MRT tests at *p* = 0.05.

**Table 5 insects-14-00445-t005:** Effect of irradiation with high daily doses of UV-B on the reproductive parameters of adult females.

Reproductive Parameter	UV-B Dose		*p*
	0 kJ m^−2^ d^−1^	57.6 kJ m^−2^ d^−1^	
Pre-oviposition period	28.39	19.86	<0.0001
Oviposition period	10.67	4.00	0.0173
Post-oviposition period	6.94	9.14	0.6687

## Data Availability

The statements, opinions, and data contained in all publications are solely those of the individual author(s) and contributor(s) and not of MDPI and/or the editor(s). MDPI and/or the editor(s) disclaim responsibility for any injury to people or property resulting from any ideas, methods, instructions, or products referred to in the content.

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
