# Peer review of "Susceptibility of Diaphorina citri to Irradiation with UV-A and UV-B and the Applicability of the Bunsen–Roscoe Reciprocity Law"

_insects, 2023, doi:10.3390/insects14050445_

Round 1

Reviewer 1 Report

I have carried out review of the article “Susceptibility of Diaphorina citri to irradiation with UV-A and UV-B and the applicability of the Bunsen-Roscoe reciprocity law.” by Parajuli et al.

I understand that the authors built up their research as a follow-up to a previously conducted study that argued several environmental factors believed to moderate psyllid populations were not responsible, thereby predicting ultraviolet radiation as the major effector at locations above 1200 m asl. Here, the current study tested this hypothesis.

The authors claimed, citing a previous paper in L16-18 that, NONE of RH, rainfall, temperature, predators, parasitoids, and entomopathogens, but ultraviolet radiation, was responsible for mediating psyllids populations at 1200m asl” Is this really correct? Does this mean, the cited paper examined all aforementioned biotic factors at 1200 m asl and found no effects on psyllid populations?

Simply put, if the reference article does not investigate the effects of all of these abiotic and biotic factors, I advise the authors to avoid overstating and modify this statement as appropriate. The authors, in fact, place too much emphasis on this statement, which was first mentioned in the Simple Summary, and again in the Abstract.

Aside from this and a few other minor errors with grammar, expression, and general writing, as I have highlighted in some parts below, the study is generally well-structured and conducted. The manuscript could be accepted if the authors are able to make the necessary changes.

L16 – ‘,’ after parasitoids

L16-18 – Check again and correct this statement, as suggested above.

L16 - ‘ambient temperature’ not ‘ambient temperatures'

L20-21 – “Immature stages were less affected when adults were treated with UV light.” What is the meaning of this statement? Adults were treated with UV, but immature stages were less affected? Kindly, amend.

L25 – Where is this ‘ Bushan’ even? Provide more details, i.e. province, and name of the country, as appropriate.

L40 – The authors might consider revising the keywords, as some of the keywords: “Diaphorina citri; UV-A; UV-B”, are already contained in the paper title.

L92 – Authors might consider providing details on location of collection of natural colonies from the field, and the time of collection. Also, indicate how long the psyllids were reared in the laboratory before the commencement of the research.

L111 – Provide details of manufacturer and country of production for stereo microscope.

L108, L128 – I am just wondering what the logic behind the disparity in the number of mature adults introduced on the shoots for egg laying is; first it was 15 when testing ‘Effects on egg hatchability’, but when it was time to examine ‘Effects on early instar nymphs’, only 10 adults were introduced. Perhaps, it would have been more appropriate to maintain the same number across each experiment.

In addition, I am just wondering if the authors deemed it necessary to identify the sexes of the 15 or 10 adult psyllids maintained on the new shoot for egg laying, as they did not mention this in the manuscript. This would have been a technical mistake if they had failed to do so, as it would definitely have influenced the outcome of the study. What if 8 out of the 10 adults introduced were male, or maybe 4 or 2 in another?"

Then again, I assume, perhaps the main objective was to obtain the cumulative number of eggs, and not necessarily to identify the number of eggs laid per shoot?"

L130 – Please be consistent in your writing ‘stereo microscope’ on some occasion, ‘stereomicroscope’ in another.

L131 – ‘Immediately thereafter,’ ?? Please, revise this grammar blunder.

L150 – I can see the identification of adult sexes was mentioned here for the first time. Also, should do the same as I have highlighted above.

Author Response

Point 1: The authors claimed, citing a previous paper in L16-18 that, NONE of RH, rainfall, temperature, predators, parasitoids, and entomopathogens, but ultraviolet radiation, was responsible for mediating psyllids populations at 1200m asl” Is this really correct? Does this mean, the cited paper examined all aforementioned biotic factors at 1200 m asl and found no effects on psyllid populations?

Response 1: In the paper by Om et al., we did not claim that these factors are not responsible for the distribution of ACP.  Rather, we stated that there was little evidence for these factors affecting the insect’s distribution, and conditions above 1100 mASL should have allowed populations of ACP to become established.  The strength of the evidence is discussed at length in this peer-reviewed paper.  As a consequence, we hypothesised that other factors may be responsible.  As Bhutan is in a UV hotspot, we surmised that this factor should be examined and this led to the current study.

Point 2: Simply put, if the reference article does not investigate the effects of all of these abiotic and biotic factors, I advise the authors to avoid overstating and modify this statement as appropriate. The authors, in fact, place too much emphasis on this statement, which was first mentioned in the Simple Summary, and again in the Abstract

Response 2: In light of the comments made above, we have amended the text so that the strength of the evidence is more accurately described and not overstated.

Point 3: L16 – ‘,’ after parasitoids

Response 3: Accepeted – a comma has been added

Point 4: L16 - ‘ambient temperature’ not ‘ambient temperatures'

Response 4: Accepted – ‘temeratures’ changed to temperature’

Point 5: L20-21 – “Immature stages were less affected when adults were treated with UV light.” What is the meaning of this statement? Adults were treated with UV, but immature stages were less affected? Kindly, amend.

Response 5: Here, immature stages mean, the F1 generation egg and nymphs produced by adults that were directly exposed to UV-A and UV-B light.  The sentence has been amended to state this effect.

Point 6: L25 Where is this ‘Bushan’ even? Provide more details, i.e. province, and name of the country, as appropriate.

Response 6: Bhusan should be Bhutan. It is written as Bhutan in the WORD version of the manuscript and must have become changed when the PDF was produced.

Point 7: L40 – The authors might consider revising the keywords, as some of the keywords: “Diaphorina citri; UV-A; UV-B”, are already contained in the paper title.

Response 7: The selection of keywords has been changed.

Point 8: L92 – Authors might consider providing details on location of collection of natural colonies from the field, and the time of collection. Also, indicate how long the psyllids were reared in the laboratory before the commencement of the research.

Response 8: This information has been added to the ms.

Point 9: L111 – Provide details of manufacturer and country of production for stereo microscope.

Response 9: This information has been added to the ms.

Point 10: L108, L128 – I am just wondering what the logic behind the disparity in the number of mature adults introduced on the shoots for egg laying is; first it was 15 when testing ‘Effects on egg hatchability’, but when it was time to examine ‘Effects on early instar nymphs’, only 10 adults were introduced. Perhaps, it would have been more appropriate to maintain the same number across each experiment.

Response 10: We used males and females in a 2:1 ratio to get more eggs (we assumed higher copulation helps to lay more eggs). And for, nymphs, the number of adults was decreased to 10.  In our preliminary experiment, when we used 15 adults for experiments on nymphs, the eggs laid and hatching % was very high.  Consequently, the mortality of nymphs was also high in both control and treatment due to overpopulation. When we reduced the adult number, the eggs laid was reduced and it was okay to continue the experiment.

Point 11: Then again, I assume, perhaps the main objective was to obtain the cumulative number of eggs, and not necessarily to identify the number of eggs laid per shoot?

Response 11: Yes, we wished to determine the cumulative number of eggs.

Point 12:  L130 – Please be consistent in your writing ‘stereo microscope’ on some occasion, ‘stereomicroscope’ in another.

Response 12: Stereomicrocope is now used throughout the manuscript.

Point 13:  L131 – ‘Immediately thereafter,’ ?? Please, revise this grammar blunder.

Response 13: The sentence has been amended.

Point 14:  L150 – I can see the identification of adult sexes was mentioned here for the first time. Also, should do the same as I have highlighted above.

Response 14: We do not understand the referee’s comments.  The identification of the sex of adults is mentioned here so that they could be put in male-female pairs.

Reviewer 2 Report

This manuscript by Parajuli et al. presents an interesting and important contribution to the relatively understudied effects of UV radiation on insect fitness. It focuses on the Asian citrus psyllid (Diaphorina citri), an economically important pest species that had not previously been investigated in that matter.

In this laboratory study, the authors examine multiple fitness parameters throughout this psyllid’s life stages and sexes, even including the F1 generation, providing a thorough picture of its susceptibility to UV damage. More specifically, they investigated the effect of different doses of UV-A and UV-B on the survival of eggs, nymphs, and adults, on the length of the pre-oviposition period, of the oviposition period, of the post-oviposition period, on female fecundity, and on the survival and development time of the F1 generation (carryover effects).

The insects were exposed to different UV radiation doses for different time periods over consecutive days, which generally had negative effects on fitness that increased according to the UV dose. Notably, there was a reduction in survival, especially during the nymph stage, as well as of the longevity and fecundity of adult females. As should have been expected, UV-B exposure was more damaging than UV-A exposure.

I have 6 general comments on the manuscript that are necessary to address:

1. UV radiation is referred to in different ways and using different abbreviations throughout the manuscript; I suggest standardization. “UV” is specified both as an abbreviation of “ultraviolet light” (line 26), but also of “ultraviolet” (line 43). Meanwhile, “UVR” is specified as an abbreviation of “ultraviolet radiation”. To be more accurate and as to not confuse the reader, I suggest referring to ultraviolet as either a form of “light” or of “radiation”, since these words are not exactly synonyms. UV “radiation” is preferred in photobiology, since “light” most often refers to wavelengths that are visible to humans. Some of my suggestions in the more specific comments and edits below reflect that position.

2. The manuscript would benefit from detailing (either in the introduction or discussion sections) why UV radiation is typically separated in three wavebands (difference in energy per photon, UV vision) as well as the physiological (or behavioural) effects that are attributed (or expected to occur) specifically to UV-A or UV-B exposure. The fact that UV-B was found to be more damaging than UV-A (even at lower doses) in this paper is not surprising given its higher energy per photon and direct photochemical effects, and this should be clearly stated.

3. More details should be available on the lamps and setups (rearing, experimental) in the Materials and Methods section (Insect culture and UV light subsections). Information such as what type of lamp were used in the rearing chambers and to create the white light (LED, fluorescent?), at what distance from the lamps were the insects placed, at what times of day were the psyllids exposed to UVR, and what were the models of the UV meters and UV lamps used, would be important to better understand the setup and allow for reproduction of the experiment. Ideally, we would know the spectral output (or at least the wavelength range) of the LED lamps as well as the white light and the general light environment in the insect cultures.

4. Some interesting parts of the discussion should be expanded on. Notably, photoreactivation is mentioned twice (lines 508, 564) as a phenomenon that mitigates UV damage, but the reason as to why should be explained further as this term is not generally known among entomologists that are not specialized in photobiology. Also, please discuss if there could be a potential synergetic effect of UV radiation and temperature on psyllid fitness. Finally, could UV radiation be used as an integrated pest management tool against the Asian citrus psyllid?

5. Some collections of facts are simply listed without being well integrated with the findings of this study, notably those contained in the last two paragraphs of the discussion. Importantly, comparisons between the numerous UV doses used in the experiments and those reported from different localities and elevations should be more extensive and brought up earlier, perhaps in the Materials and Methods section. Throughout the manuscript, it should be clear whether your experimental UV doses can be considered as “realistic” for this pest.

6. Some information is lacking from the “Data analysis” section, namely specifying how the data not related to survival (e.g., reproduction, fecundity) was analysed and how were the assumptions for the ANOVAs (normally distributed errors, homoscedasticity) and regressions checked. As the multiple comparison test used (Duncan’s MRT) is generally considered to be too liberal, more prone to false positives than notably the Tukey HSD test, I am also wondering if other types of tests would have given the same results. Finally, why was Kaplan-Meier survival analysis only applied to nymph survival, and not adult longevity? A figure showing Kaplan-Meier survival curves for adult males and females would be more informative than the bar graphs in Fig.6.

More specific comments and suggested edits follow:

Lines 43-45: I think the next sentence (lines 45-46) would make a better start for your paper, providing context as to why you are studying UV radiation instead of just stating general facts about its wavelength ranges. These could then be brought up to introduce the difference between their specific effects (thereby explaining why the distinction is important). Also, the wavebands mentioned do not match with those of the following reference [1], which are 200-280 nm for UV-C, 280-315 nm for UV-B, and 315-400 nm to UV-A. Using these wavebands would be more appropriate since they are notably used by the International Commission on Illumination.

Line 18: Add “(UVR)” after “ultraviolet radiation”.

Lines 21 & 65: Replace “UV light” with “UVR”.

Line 26: Replace “ultraviolet light (UV)” with “ultraviolet radiation (UVR)”.

Lines 45-46: Adding another reference would help support this broad statement, as the one provided is solely on human health.

Lines 48-50: This sentence is unclear to me; I suggest you rewrite it. Do you mean that some insects can benefit indirectly from the damaging effects of UV radiation on their natural enemies?

Line 55: Replace “radiation of waveband of UVR” with “it”.

Line 57: Specify “longer UV wavelengths”.

Line 58: The Asian citrus psyllid is abbreviated as “ACP”, but that abbreviation is only used once more in the manuscript (line 63). The insect is thereafter often referred to as “the psyllid”, which is fine in the context of the Material and Methods section for instance, but less so in the Introduction and Discussion, where it makes it notably hard to know if other psyllid species than the ACP are included when “psyllid populations” or “psyllids” are mentioned. In those sections, the more precise “ACP”, “D. citri”, or “Asian citrus psyllid” should be used.

Line 69: The abbreviation “ASL” should be indicated at least once to mean “above sea level”.

Line 75: The word “that” is doubled.

Line 80: Remove “the light of” and “Therefore,”.

Line 96: Replace “light” with “radiation”.

Line 114: Replace “UV light” with “UVR”.

Lines 132, 133, 135 & 156: Remove “light” to standardize how you refer to UV-A and UV-B (or add “radiation” to every instance throughout the manuscript).

Lines 134-135: Whenever (experimental) UV doses are mentioned, it would help to present them as daily UV doses instead of only total doses, especially to compare between treatments and measurements in natural conditions.

Lines 138, 162 & 172: Replace “UV light” with “UVR”.

Line 150: How exactly are they sexed based on their morphology (which traits)?

Lines 151, 177, 289, 342, 366, 399, 439 & 443: Remove “radiation” to standardize how you refer to UV-A and UV-B (or add it to every instance throughout the manuscript).

Lines 157-158: Why were the last two exposure time treatments for UV-B only?

Line 166: “Longevity” seems more appropriate than “survival” for this adult fitness parameter.

Line 172: State which carryover effects were studied here.

Line 199: The part “with differences among means determined” is repetitive.

Lines 205 & 447: Replace “UV” with “UVR”.

Lines 230, 244, 261, 269, 297 & 315: In the figure legends, the replicates are indicated as n = 4, but the number of individual eggs or nymphs examined in each should also be clearly indicated here.

Lines 231, 246, 263, 271, 298, 316, 335, 356, 358, 364, 377, 419 & 426: I think you mean “Duncan’s multiple range test” as indicated in the “Data analysis section”, rather than (Tukey Honestly Significant Difference) “HSD” test.

Lines 447-449: This sentence suggests that there are no exceptions among arthropods (terrestrial, aquatic?) regarding susceptibility to UV damage. Is this really true?

Line 449: Remove “irradiation”.

Line 455: Here would be a good place to mention that UV-C doesn’t occur naturally.

Line 466: Do you mean “resulted” instead of “resulting”? The meaning of this sentence is not clear to me. Also, please specify if this is an observation from the experiments detailed in the manuscript.

Line 470: Please specify what you mean by “affected” (visible physical damage, survival?).

Line 479: This part of the sentence is unclear. In this study [46], the stink bug nymphs that emerged from darker eggs exposed to UV-A radiation retained less carryover effects than those that emerged from lighter eggs (the nymphs were not exposed to UVR).

Line 480: The second mention of reference [46] is wrong; this paper did not work on springtails.

Line 499: Add a reference at the end of this sentence.

Lines 509-510: Add a reference at the end of this sentence.

Line 553: “UVA” should be hyphenated.

Lines 563-565: The difference and link between the two parts of this sentence is unclear to me, they seem repetitive.

Lines 566-568: Are Asian citrus psyllid expected to be exposed to direct sunlight (and if so, when)? Since eggs are laid in citrus shoots (as mentioned in line 63), could we expect that they are completely shielded from UV exposure, as plant leaves usually absorb those photons? If nymphs and adults remain on leaf underside (as mentioned in line 64, a form of UV avoidance?) how do your experimental UV doses relate to realistic UV doses based on the percentages of diffuse UV radiation that are mentioned?

Line 574: “UVR radiation” should be either “UVR” or “UV radiation”.

Line 575: Are these daily UV doses, like before?

Line 581: Replace “psyllid” with “the ACP”.

Line 582: The abbreviation “HLB” is found only this once in the manuscript and is never specified to mean “huanglongbing”.

In general, I invite the authors to carefully review the entire text to improve the fluidity and clarity of the writing. Many of the edits I suggest in the specific comments and suggestions for authors should help in that matter (I have spotted several mistakes and inconsistencies) but will be sufficient to bring the manuscript to its highest potential. With so many fitness parameters examined, and UV doses used, it is hard for the reader to keep in mind everything that was done, and so it is especially important to use more standardized vocabulary when referring to specific experiments or when comparing one to another.

Author Response

  1. UV radiation is referred to in different ways and using different abbreviations throughout the manuscript; I suggest standardization. “UV” is specified both as an abbreviation of “ultraviolet light” (line 26), but also of “ultraviolet” (line 43). Meanwhile, “UVR” is specified as an abbreviation of “ultraviolet radiation”. To be more accurate and as to not confuse the reader, I suggest referring to ultraviolet as either a form of “light” or of “radiation”, since these words are not exactly synonyms. UV “radiation” is preferred in photobiology, since “light” most often refers to wavelengths that are visible to humans. Some of my suggestions in the more specific comments and edits below reflect that position.

Response. We have amended the ms so that the term ‘UV radiation ‘is used throughout.

  1. The manuscript would benefit from detailing (either in the introduction or discussion sections) why UV radiation is typically separated in three wavebands (difference in energy per photon, UV vision) as well as the physiological (or behavioural) effects that are attributed (or expected to occur) specifically to UV-A or UV-B exposure. The fact that UV-B was found to be more damaging than UV-A (even at lower doses) in this paper is not surprising given its higher energy per photon and direct photochemical effects, and this should be clearly stated.

Response. Text explaining the division into three wave bands has been added to the Introduction.

  1. More details should be available on the lamps and setups (rearing, experimental) in the Materials and Methods section (Insect culture and UV light subsections). Information such as what type of lamp were used in the rearing chambers and to create the white light (LED, fluorescent?), at what distance from the lamps were the insects placed, at what times of day were the psyllids exposed to UVR, and what were the models of the UV meters and UV lamps used, would be important to better understand the setup and allow for reproduction of the experiment. Ideally, we would know the spectral output (or at least the wavelength range) of the LED lamps as well as the white light and the general light environment in the insect cultures.

Response. Further details of the experimental setup have been added into the M&M section.  The UV meters do not have model numbers.

  1. Some interesting parts of the discussion should be expanded on. Notably, photoreactivation is mentioned twice (lines 508, 564) as a phenomenon that mitigates UV damage, but the reason as to why should be explained further as this term is not generally known among entomologists that are not specialized in photobiology. Also, please discuss if there could be a potential synergetic effect of UV radiation and temperature on psyllid fitness. Finally, could UV radiation be used as an integrated pest management tool against the Asian citrus psyllid?

Response. The phenomenon of photoreactivation has been described in the Discussion. There are interactions between UVR and temperature; higher temperatures moderate the effects of UVR. However, the ms has not examined this interaction and any discussion relating it to our results would be highly speculative. Therefore, we have not included it.  The practical relevance of this study in relation to pest control has been added to the end of the Discussion.

  1. Some collections of facts are simply listed without being well integrated with the findings of this study, notably those contained in the last two paragraphs of the discussion. Importantly, comparisons between the numerous UV doses used in the experiments and those reported from different localities and elevations should be more extensive and brought up earlier, perhaps in the Materials and Methods section. Throughout the manuscript, it should be clear whether your experimental UV doses can be considered as “realistic” for this pest.

Response. The study was carried out to determine the doses of UV-A and UV-B that affect the survival and development of D. citri. The magnitudes of the determined doses in relation to natural conditions that the psyllid might encounter are presented in the Discussion. The text in this section has been amended to better indicate that the determined doses are realistic and could affect the distribution of the pest. We have left the discussion of the relationship of the derived ED50 values in the Discussion section, as we consider it to be a discussion point.

  1. Some information is lacking from the “Data analysis” section, namely specifying how the data not related to survival (e.g., reproduction, fecundity) was analysed and how were the assumptions for the ANOVAs (normally distributed errors, homoscedasticity) and regressions checked. As the multiple comparison test used (Duncan’s MRT) is generally considered to be too liberal, more prone to false positives than notably the Tukey HSD test, I am also wondering if other types of tests would have given the same results. Finally, why was Kaplan-Meier survival analysis only applied to nymph survival, and not adult longevity? A figure showing Kaplan-Meier survival curves for adult males and females would be more informative than the bar graphs in Fig.6.

Response. Further details of the statistical procedures have been added to the M&M section. Both DMR tests and Tukey’s test were used to examine the data; there was little difference between them. To my knowledge, Kaplan-Meier analysis is used to study the drop in survival over time of a population and not applicable to survival times. The K-M analysis gives a mean survival time and the mean survival times calculated from the individual repetition were used to analyse differences in longevity among treatments.

More specific comments and suggested edits follow:

Lines 43-45: I think the next sentence (lines 45-46) would make a better start for your paper, providing context as to why you are studying UV radiation instead of just stating general facts about its wavelength ranges. These could then be brought up to introduce the difference between their specific effects (thereby explaining why the distinction is important). Also, the wavebands mentioned do not match with those of the following reference [1], which are 200-280 nm for UV-C, 280-315 nm for UV-B, and 315-400 nm to UV-A. Using these wavebands would be more appropriate since they are notably used by the International Commission on Illumination.

Response. The Introduction has been amended as suggested and the bounds of the wavebands amended.

Line 18: Add “(UVR)” after “ultraviolet radiation”. amended

Lines 21 & 65: Replace “UV light” with “UVR”. amended

Line 26: Replace “ultraviolet light (UV)” with “ultraviolet radiation (UVR)”. amended

Lines 45-46: Adding another reference would help support this broad statement, as the one provided is solely on human health.

Response. Further references have been added to support this contention.

Lines 48-50: This sentence is unclear to me; I suggest you rewrite it. Do you mean that some insects can benefit indirectly from the damaging effects of UV radiation on their natural enemies?

Response. Yes, beneficial effects of exposure to UVR can occur. The sentence has been amended to more clearly articulate this.

Line 55: Replace “radiation of waveband of UVR” with “it”.

Response.  This sentence has been amended

Line 57: Specify “longer UV wavelengths”.

Response. Information added

Line 58: The Asian citrus psyllid is abbreviated as “ACP”, but that abbreviation is only used once more in the manuscript (line 63). The insect is thereafter often referred to as “the psyllid”, which is fine in the context of the Material and Methods section for instance, but less so in the Introduction and Discussion, where it makes it notably hard to know if other psyllid species than the ACP are included when “psyllid populations” or “psyllids” are mentioned. In those sections, the more precise “ACP”, “D. citri”, or “Asian citrus psyllid” should be used.

Response. The ms has been amended to clarify when we are referring to D. citri.

Line 69: The abbreviation “ASL” should be indicated at least once to mean “above sea level”.

Response. ASL has been defined on first usage

Line 75: The word “that” is doubled.

Response. The extra ‘that’ has been removed

Line 80: Remove “the light of” and “Therefore,”.

Response.  The line has been amended

Line 96: Replace “light” with “radiation”. amended

Line 114: Replace “UV light” with “UVR”. amended

Lines 132, 133, 135 & 156: Remove “light” to standardize how you refer to UV-A and UV-B (or add “radiation” to every instance throughout the manuscript). amended

Lines 134-135: Whenever (experimental) UV doses are mentioned, it would help to present them as daily UV doses instead of only total doses, especially to compare between treatments and measurements in natural conditions.

Response. All doses are daily doses. This has been made clear throughout the ms.

Lines 138, 162 & 172: Replace “UV light” with “UVR”. amended

Line 150: How exactly are they sexed based on their morphology (which traits)?

Response. The sexes are readily identified as the genitalia cause large differences in the shape of the tip of the abdomen. This is easily detected using a stereomicroscope.

Lines 151, 177, 289, 342, 366, 399, 439 & 443: Remove “radiation” to standardize how you refer to UV-A and UV-B (or add it to every instance throughout the manuscript). amended

Lines 157-158: Why were the last two exposure time treatments for UV-B only?

Response. The student had limited time and resources available to her.

Line 166: “Longevity” seems more appropriate than “survival” for this adult fitness parameter.

Response. This change has been made.

Line 172: State which carryover effects were studied here.

Response. Carry over effects defined

Line 199: The part “with differences among means determined” is repetitive.

Response. Repetition removed

Lines 205 & 447: Replace “UV” with “UVR”. amended

Lines 230, 244, 261, 269, 297 & 315: In the figure legends, the replicates are indicated as n = 4, but the number of individual eggs or nymphs examined in each should also be clearly indicated here.

Response. Here n=4 means the replication per treatment. The total number of eggs and nymphs directly exposed to UV-A and UV-B radiation and the number of nymphs hatched from eggs after direct UV exposure are given in the manuscript. (Page 6-13)

Lines 231, 246, 263, 271, 298, 316, 335, 356, 358, 364, 377, 419 & 426: I think you mean “Duncan’s multiple range test” as indicated in the “Data analysis section”, rather than (Tukey Honestly Significant Difference) “HSD” test.

Response.  Change made throughout ms

Lines 447-449: This sentence suggests that there are no exceptions among arthropods (terrestrial, aquatic?) regarding susceptibility to UV damage. Is this really true?

Response. Yes, true at least at some dose. We use ‘can be’ to reflect this.

Line 449: Remove “irradiation”.

Response. ‘irradiation’ removed

Line 455: Here would be a good place to mention that UV-C doesn’t occur naturally.

Response. This is detailed in the Introduction

Line 466: Do you mean “resulted” instead of “resulting”? The meaning of this sentence is not clear to me. Also, please specify if this is an observation from the experiments detailed in the manuscript.

Response. Change made

Line 470: Please specify what you mean by “affected” (visible physical damage, survival?).

Response. Information given

Line 479: This part of the sentence is unclear. In this study [46], the stink bug nymphs that emerged from darker eggs exposed to UV-A radiation retained less carryover effects than those that emerged from lighter eggs (the nymphs were not exposed to UVR).

Response. The second part of this sentence has been amended

Line 480: The second mention of reference [46] is wrong; this paper did not work on springtails.

Response. The reference has been changed to Leinaas 2002

Line 499: Add a reference at the end of this sentence.

Response. No reference can be provided as this is a conjecture by the authors

Lines 509-510: Add a reference at the end of this sentence.

Reference added

Line 553: “UVA” should be hyphenated.

Response. Hyphen added

Lines 563-565: The difference and link between the two parts of this sentence is unclear to me, they seem repetitive.

Response. This section on the ms has been amended

Lines 566-568: Are Asian citrus psyllid expected to be exposed to direct sunlight (and if so, when)? Since eggs are laid in citrus shoots (as mentioned in line 63), could we expect that they are completely shielded from UV exposure, as plant leaves usually absorb those photons? If nymphs and adults remain on leaf underside (as mentioned in line 64, a form of UV avoidance?) how do your experimental UV doses relate to realistic UV doses based on the percentages of diffuse UV radiation that are mentioned?

Response. The discussion details how diffuse UV-A and UV-B radiation exist in plant canopies to which the psyllid would be exposed.

Line 574: “UVR radiation” should be either “UVR” or “UV radiation”.

Response. Amended – now UV radiation

Line 575: Are these daily UV doses, like before?

Response. This is now clarified

Line 581: Replace “psyllid” with “the ACP”.

Response. Psyllid is now replaced with D. citri

Line 582: The abbreviation “HLB” is found only this once in the manuscript and is never specified to mean “huanglongbing”.

Response. HLB is now in full form

Comments on the Quality of English Language

In general, I invite the authors to carefully review the entire text to improve the fluidity and clarity of the writing. Many of the edits I suggest in the specific comments and suggestions for authors should help in that matter (I have spotted several mistakes and inconsistencies) but will be sufficient to bring the manuscript to its highest potential. With so many fitness parameters examined, and UV doses used, it is hard for the reader to keep in mind everything that was done, and so it is especially important to use more standardized vocabulary when referring to specific experiments or when comparing one to another.

Response. The comments of the reviewer have, indeed, improved the flow and expression within the ms. The authors have reread the ms and are satisfied with its content and presentation.
